# Influence of adiposity and sex on SARS-CoV-2 antibody response in vaccinated university students: A cross-sectional ESFUERSO study

Adriana L. Perales-Torres[1], Lucia M. Perez-Navarro[2], Esperanza M. Garcia-Oropesa[1], Alvaro Diaz-Badillo[3,4], Yoscelina Estrella Martinez-Lopez[5], Marisol Rosas[1], Octelina Castillo[1], Laura Ramirez-Quintanilla[1], Jacquelynne Cervantes[6], Edda Sciutto[7], Claudia X. Munguia Cisneros[8], Carlos Ramirez-Pfeiffer[1,4], Leonel Vela[9], Beatriz Tapia[9], Juan C. Lopez-Alvarenga[4,9] *

1 Unidad Académica Multidisciplinaria Reynosa Aztlan, Universidad Autónoma de Tamaulipas, Reynosa, Tamaulipas, México, 2 Departamento de Nefrología, Hospital General de México Dr. Eduardo Liceaga, Mexico City, Mexico, 3 Public Health Research Group, Department of Life Sciences, Texas A&M University-San Antonio, San Antonio, Texas, United States of America, 4 Escuela de Medicina, Universidad México Americana del Norte, Reynosa, Tamaulipas, Mexico, 5 School of Public Health, University of Texas Health Science Center at Houston, Brownsville, Texas, United States of America, 6 Facultad de Medicina Veterinaria y Zootecnia, Universidad Nacional Autónoma de México, Mexico City, Mexico, 7 Instituto de Investigaciones Biomedicas, Universidad Nacional Autónoma de México, Mexico City, Mexico, 8 Centro Especializado de Diabetes y Metabolismo CEDIAMET, Universidad Mexico Americana del Norte, Reynosa, Tamaulipas, Mexico, 9 School of Medicine, University of Texas Rio Grande Valley, UTRGV, Edinburg, Texas, United States of America

* juan.lopezalvarenga@utrgve.edu

## Abstract

Prior studies have identified various determinants of differential immune responses to COVID-19. This study focused on the Ig-G anti-RBD marker, analyzing its potential correlations with sex, vaccine type, body fat percentage, metabolic risk, perceived stress, and previous COVID-19 exposure. In this study, data (available in S1 Data) were obtained from 108 participants from the ESFUERSO cohort, who completed questionnaires detailing their COVID-19 experiences and stress levels assessed through the SISCO scale. IgG anti-RBD concentrations were quantified using an ELISA assay developed by UNAM. Multiple regression analysis was employed to control for covariates, including sex, age, body fat percentage, body mass index (BMI), and perceived stress. This sample comprised young individuals (average age of 21.4 years), primarily consisting of females (70%), with a substantial proportion reporting a family history of diabetes, hypertension, or obesity. Most students had received the Moderna or Pfizer vaccines, and 91% displayed a positive anti-RBD response. A noteworthy finding was the interaction between body fat percentage and sex. In males, increased adiposity was associated with decreased Ig-G anti-RBD concentration; in females, the response increased. Importantly, this pattern remained consistent regardless of the vaccine received. No significant associations were observed for dietary habits or perceived stress variables. This research reports the impact of sex and body fat percentage on the immune response through Ig-G anti-RBD levels to COVID-19 vaccines. The implications of these findings offer a foundation for educational initiatives and the formulation of preventive policies aimed at mitigating health disparities.

**Data Availability Statement:** The dataset is available at https://scholarworks.utrgv.edu/cgi/ir_submit.cgi?context=som_pub.

**Funding:** This study was supported by a grant for ALPT from COTACYT-2021-01-23, Mexico. The funders had no role in study design, data collection and analysis, decision to publish, or preparationof the manuscript.

**Competing interests:** The authors have declared that no competing interests exist.

## Introduction

Sex differences in metabolism and insulin resistance have been demonstrated in our previous studies with children [1–3]. Similar findings from various cultural and social contexts support differences in insulin resistance and metabolic-associated liver disease in adults [4]. These differences are partly due to the protective effect of endogenous estrogens on various tissues, including brain, liver, skeletal muscle, adipose tissue, and pancreatic beta cells [5], and other underlying mechanism such as AMPdependent protein kinase (AMPK) activation [6].

Immune responses influenced by sex differences, particularly in LTR8, were reported during the COVID-19 pandemic [7]. Other studies have investigated factors that impact the serum levels of antibodies produced by the COVID-19 vaccine [8–10]. Age is a critical factor that plays a significant role in determining the immune response. Elderly individuals with obesity and non-prior infection had reduced antibody titers against SARS-CoV-2 spike antigen after CoronaVac vaccine (manufactured in China) compared to non-obese people [11]. Lower antibody response after receiving two doses of the Pfizer vaccine has also been linked to central obesity (correlation of $r = -0.3$), the presence of hypertension, and smoking habits, with no notable differences by gender [12].

During the pandemic, it became evident that metabolic imbalances associated with obesity could increase the severity of COVID-19 and the risk of mortality [13, 14]. Diet can induce metabolic and immune impairments, which may vary based on sex, as shown in animal studies [15, 16]. Interestingly, losing weight has been shown to improve the adaptive immune response, particularly an increase in INF-g2 levels following the administration of two doses of mRNA vaccine [17].

Sex and body weight interaction can also result in varying immune responses. For individuals with a BMI $>40$ kg/m$^2$, there were no discernible differences in IgG antibody levels between the sexes [18]. In contrast, those with normal weight showed higher levels among males [19]. Physical activity and migration have influenced immunological markers and health outcomes [20, 21]. Psychological stress is another recognized variable affecting the immune response [22, 23] Early-life adversity, affecting 39% of the world's population, has been associated with neuroinflammation and increased levels of soluble tumor necrosis factor in animal studies [24]. Diet and stress can serve as effect modifiers of biological variables like sex and adiposity.

The present study focused on a nested sample of students from the ESFUERSO (**ES**tudio de la **F**rontera **U**rbana para las **E**nfe**R**medades y factores a**S**ociados a la **O**besidad) program. These students experienced the impact of the COVID-19 pandemic, which forced them into home confinement and led to changes in their habits.

The development of mRNA vaccines has been a long journey, beginning with cell cultures and laboratory animal testing, followed by industry investment and veterinary applications, ultimately leading to human vaccination in 2020. These efforts have been ongoing for three decades [25].

This study aimed to analyze the mathematical function of the immune response to the receptor-binding domain (RBD), a protective epitope found in the S protein [26], by examining the Ig-G response among young students from Mexico living near the US-Mexico border. The analyzed factors included sex, body fat, psychological stress, and food consumption (Fig 1A).

## Methods

### Study sample

In 2018, the ESFUERSO cohort study was initiated, focusing on 500 first-year students from two universities in Reynosa, Tamaulipas. Characteristics of this sample were described

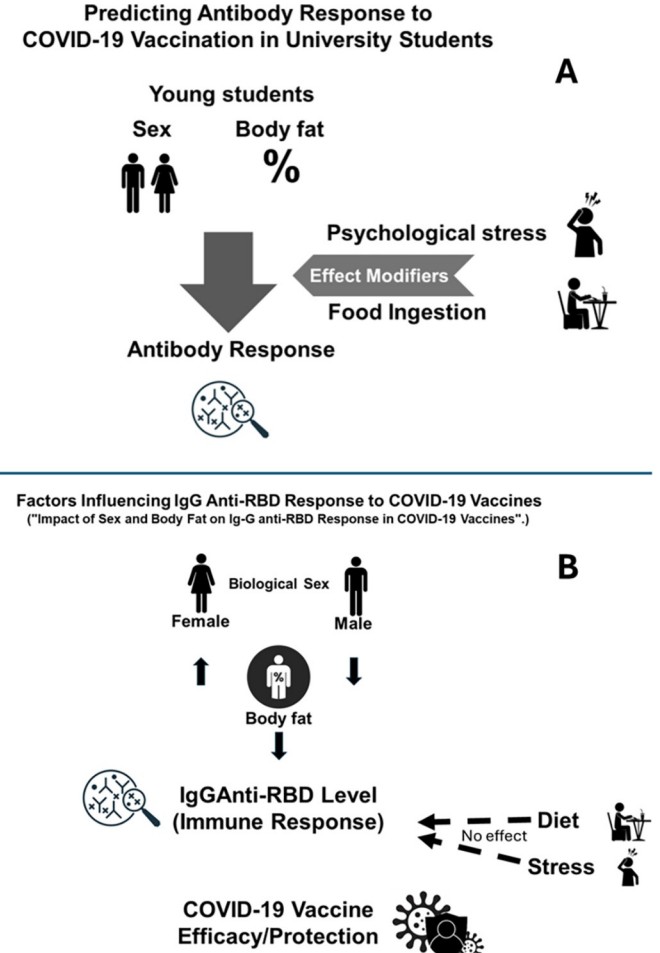

**Fig 1.** The top panel (A) presents the initial hypothesis, in which we expected sex and body fat to affect the immune response to COVID-19 vaccination, with psychological factors and food ingestion acting as effect modifiers. The bottom panel (B) concludes that the immune response differed between males (trend to negative association) and females (trend to positive association), and the effect modifiers did not play a role in this study.

elsewhere [27]. In summary, 70% of the participants were considered to have metabolic risk if they had a BMI > 30 or a family history of obesity, diabetes, or hypertension.

For the current study, we intended a proportional sample with 70% of students with metabolic risk who accepted to participate and could attend near the university facilities. Due to the pandemic, not all students could attend, and only a subset of 116 students were contacted between September 1st and October 31st, 2021. During this period, we obtained signed informed consent (see Ethics Statement below for details), conducted questionnaire surveys, collected anthropometric measurements, and collected blood samples from 108 students in compliance with the COVID-19 protocol managed by the universities. Eight students initially agreed to participate but did not complete the surveys or clinical measurements.

## Measures

The questionnaires and methods used in the ESFUERSO study have been described elsewhere [27]. Briefly, the questionnaires collected information on family metabolic risk, anxiety, depression, and uncertainty. The Cronbach α coefficient from each question ranged from

0.72–0.96. The test-retest for agreement in categorical variables was a kappa coefficient between 0.5–0.91 and an intraclass correlation coefficient between 0.73–0.96 in continuous variables. The stress during the pandemic was evaluated with SISCO (Modelo **SIS**témico **CO**gnoscitivista para el estudio del estrés académico), to assess distress, uncertainty, lack of sleep, sadness, and anxiety. The SISCO was validated in Mexico and other Latin American countries with a Cronbach α coefficient of 0.9 with high homogeneity [28]. The food ingestion scores were calculated using a semi-quantitative questionnaire about 42 selected regional foods. The weighted kappa was greater than 0.6 for all items in a reliability study [29]. All the questionnaires were administered electronically, with students completing them on their cell phones. This approach ensured efficient data collection and minimized the need for physical paperwork or in-person administration.

Weight, height, and acanthosis nigricans grade [30] were assessed and registered at the universities by a standardized nutritionist [27]. The body fat percentage was measured by bioelectrical impedance using a body composition analyzer (Tanita TBF-300A). Blood samples were obtained between 7 to 9 am (after an overnight fasting period) for the measurement of serum concentration of Ig-G anti-RBD by indirect ELISA [26]. The samples underwent centrifugation following collection, and four aliquots were stored at -20 C. These aliquots were transported to Mexico City in cold conditions in November 2021 for antibody analysis. The effective neutralizing concentration of anti-RBD IgG was assessed, and this variable was analyzed in both continuous and dichotomous dimensions. A threshold of 1.0 for the anti-RBD IgG ratio was established to differentiate between effective and non-effective neutralization.

## Statistical analysis

Descriptive statistics included percentages for categorical variables and means with standard deviations or medians and interquartile range (Q1, Q3) for continuous variables. Inferential statistics utilized regression analysis to evaluate the relation in the SISCO questionnaire scores; the food ingestion scores with 42 selected regional foods were reduced to 12 factors (KMO = 0.64 supporting suitability analysis for principal components and varimax rotation) [29]. Locally weighted regression (lowess) was performed to assess nonlinear relationships. The analysis used antibody concentration as the dependent variable, adjusting for sex, age, metabolic risk, body fat percentage, and BMI. Multiplicative interactions for covariates were analyzed. The variance inflation factor (VIF) was calculated to evaluate multicollinearity in variables used in linear models without interactions [31]. Multicollinearity occurs when two or more independent variables in a regression model are highly correlated, making accurately estimating each variable's effect on the dependent variable difficult. The VIF quantifies the extent to which multicollinearity increases an estimated regression coefficient's variance (i.e., the uncertainty or variability) [32, 33].

We evaluated three models (presented in Table 2) covering sex, body fat, psychological variables, and food ingestion; some included an interaction term for body fat percentage and sex. The models included first—to third-grade polynomial multiple regression, and the goodness of fit was assessed using a squared error calculation. The best goodness of fit for data was achieved with polynomial multiple regression using a sample of 108 students with complete data. All analyses were performed with Stata V18.0 (StataCorp, College Station, TX).

## Ethics statement

The protocol and informed consent were approved by the Comite de Etica Institucional de la Unidad Academica Multidisciplinaria Reynosa-Aztlan (CEI-UAMRA) number registration CEI-UAMRA 005/2019/CEI under Health normativity (NOM-012-SSA-3-212). The informed

consent detailed the risks and benefits and ensured no cost to the participants, who had the right to withdraw from the project without any questions. All participants signed the approved informed consent. The present report followed the STROBE recommendations for cross-sectional studies [34].

## Results

A total of 108 students in the ESFUERSO cohort were enrolled during the 3rd year of the cohort follow-up in Reynosa. The participants had a mean age of 21.4 (SD 1.0) years, an average BMI of 27.9 (SD 6.2), and gender distribution of 69% (75 out of 108) female students. The presence of T2D, hypertension, obesity, or a combination of the conditions was identified in 70% of the students. Notably, there were no discernible differences between universities on metabolic risk, anthropometry, sex, and commercial brand of vaccine or the presence of adequate antibody levels (Table 1). Only 3 (3%) of the students from the private university had no COVID-19 vaccination at the time of the study, and they also belonged to the metabolic risk group (Pearson standardized distance >3.0). From vaccinated students, 97 (90%) were immunized with Moderna or Pfizer, and only 8 (7%) with other vaccines (Johnson & Johnson n = 1, Cansino n = 6, Sinovac n = 1). The prevalence of positive anti-RBD was 91%, with similar values by sex (Table 1).

The multicollinearity analysis showed BMI and fat percentage had VIF = 4.8 and 1/VIF = 0.02. This suggested a moderate level of collinearity, so we analyzed separating both variables to maintain a clear interpretation of the regression coefficients. The anthropometric analysis showed the body fat percentage had the lowest BIC (BIC = 259.9) compared with BMI (BIC = 261.2) and waist circumference (BIC = 264.4). Table 2 shows the results of body fat percentage with sex interaction. The body fat percentage interaction with sex was statistically significant, supporting the serum concentration of anti-RBD decreased as adiposity increased in men (p = 0.034 for a second-grade term). Still, anti-RBD increased with adiposity in women (p = 0.01, p = 0.015 for second and third-grade terms) (Fig 2). The interaction remained despite the vaccine types. The adjusted model minimized the mean squared error (Root

**Table 1. Descriptive statistics of general variables by sex.**

| Variable | Female students (n = 75) | Male students (n = 33) |
|---|---|---|
| Age (years)* | 21.4 ±1.1 | 21.4 ±0.9 |
| BMI (kg/m$^2$) * | 27.7 ± 6.1 | 28.4 ± 6.3 |
| Body fat (%)* | 33.04 ± 10.45 | 24.2 ± 10.34 |
| Waist circumference (cm) * | 83.3 ±13.2 | 91.9 ±13.7 |
| Neck circumference (cm) * | 34.4 ±3.2 | 40.4 ±10.6 |
| Systolic blood pressure (mmHg) * | 113 ±9 | 120.8 ±18.7 |
| Diastolic blood pressure (mmHg) * | 77 ±7 | 81.1 ±92.9 |
| Positive RBD (%) | 66 (88%) | 31 (94%) |
| Anguish** | 3 (2, 5) | 3 (1, 4) |
| Uncertainty** | 3 (2, 4) | 2 (1, 4) |
| Lack of sleep** | 4 (2, 5) | 2 (1, 3) |
| Sadness** | 3 (3, 5) | 2 (1, 3.5) |
| Anxiety** | 4 (3, 5) | 3 (1.5, 4) |
| Acanthosis nigricans** | 0 (0, 2) | 1 (0, 2) |

* Continuous variables presented mean and standard deviation.

**Ordinal variables present the median along the first and third quartile (Q1, Q3).

**Table 2. Statistical models test the hypothesis.**

| | Model 1. Effect of psychological variables | Model 2. Effect of food preferences | Model 3. Interaction between body fat and sex |
|---|---|---|---|
| Sex (Male) | 0.87 (-0.07, 1.81)† | 1.0 (0.05, 1.9) ‡ | -0.66 (-2.79, 1.46) † |
| Fat percentage (%) | 0.02 (0.002, 0.037)‡ | 0.02 (0.004, 0.038) ‡ | -0.23 (-0.41, -0.04) ‡ |
| Sex*Fat% (Male) | -0.03 (-0.06, 0.005) † | -0.03 (-0.06, -0.001) ‡ | 0.16 (-0.15, 0.48) |
| Sex*Fat%^2 (Female) | | | 0.01 (0.002, 0.017)*** |
| Sex*Fat%^2 (Male) | | | 0.004 (-0.01, 0.02) |
| Sex*Fat%^3 (Female) | | | -0.001 (-0.0001, -0.00002) ‡ |
| Sex*Fat%^3 (Male) | | | -0.0001 (-0.0003, 0.0001) |
| Distress | -0.05 (-0.20, 0.10) | | |
| Uncertainty | 0.11 (-0.07, 0.28) | | |
| Lack of sleep | -0.06 (-0.18, 0.07) | | |
| Sadness | 0.01 (-0.13, 0.16) | | |
| Anxiety | 0.01 (-0.14, 0.16) | | |
| Twelve factors of food | | -0.12 (-0.27, 0.03) to 0.10 (-0.04, 0.25) | |
| Adj R^2 | 0.001 | 0.04 | 0.08 |
| Root MSE | 0.76 | 0.75 | 0.74 |

The regression coefficients (95%CI) are presented in a summary of the regression models analyzed. To ensure the integrity of our analysis, we addressed potential collinearity issues (detailed in the text) and limited our examination to relationships among variables based on established clinical criteria. The table includes two key statistical measures for each model: the Adjusted $R^2$, representing the proportion of variance explained by the model (adjusted for the number of predictors), Mean Square Error to quantify the standard deviation of the residuals, and measuring the model's accuracy. Adj $R^2$: Adjusted coefficient of determination. Root MSE: Root of Mean Square Error.

†$p < 0.10$

‡$p < 0.05$

***$p < 0.01$.

MSE = 0.74) and adjusted coefficient of determination ($R^2 = 0.08$), compared with other models (Fig 3 and Table 2). The residuals adjusted to a normal distribution (Shapiro-Wilkins $p = 0.136$).

No differences due to metabolic risk factors or effective antibody concentration were found for food consumption and psychological variables (distress, uncertainty, lack of sleep, sadness, and anxiety) presented in Table 2.

## Discussion

Based on the results of our study, it appears that the neutralizing anti-RBD response to the COVID-19 vaccine is influenced by a multiplicative interaction of sex and body fat percentage. Specifically, females tend to have increased responses, while males tend to have decreased responses (Fig 3). Stress scores do not appear to have significant effects (Fig 1B).

This observation aligns with existing research on sex-based differences in immune responses. For instance, a study involving the Cameron County Hispanic Cohort, which included 624 participants with a mean age of 50 (SD 14) years, has previously reported sex-specific variations in adipokines and carotid intima-media thickness [35]. The present study extends these findings to a younger cohort, specifically individuals in the final stages of adolescence residing near the U.S.-Mexico border. This highlights the relevance of considering age and geographic location when examining immune response differences between sexes.

Another study conducted in Mexico on 980 adult participants with a median age of 50 (Q1: 36, Q3: 54) who had obesity before mass vaccination sheds light on this [36]. The authors identified independent factors associated with SARS-Cov-2 infection in a symptomatic group.

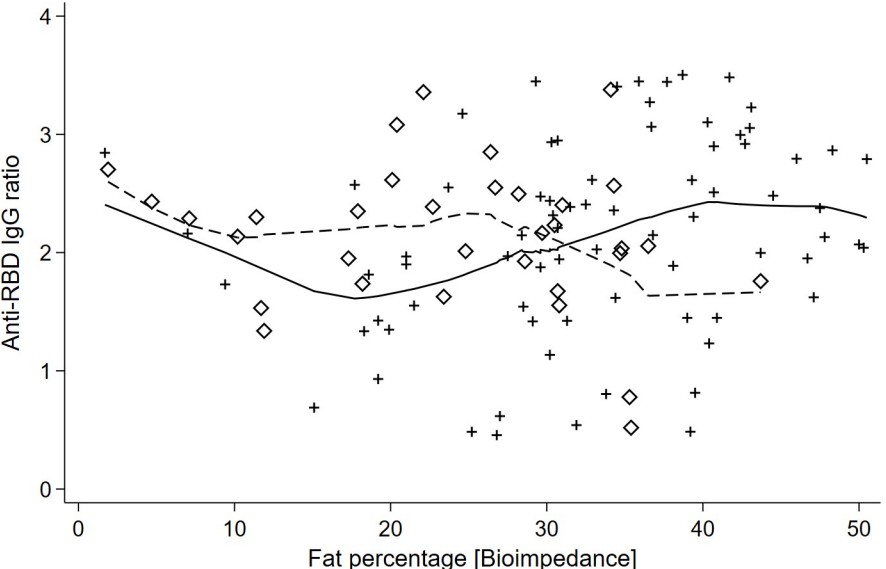

**Fig 2. Sex differences in neutralizing anti-RBD IgG ratios as a function of body fat percentage.** This figure illustrates the relationship between neutralizing anti-RBD (Receptor Binding Domain) IgG ratios and body fat percentage among university students, using the Lowess (Locally Weighted Regression) smoothing technique to highlight trends. Females and males are distinguished by diamonds and crosses, respectively, with the Lowess curve for females presented as a continuous line and for males as a dashed line.

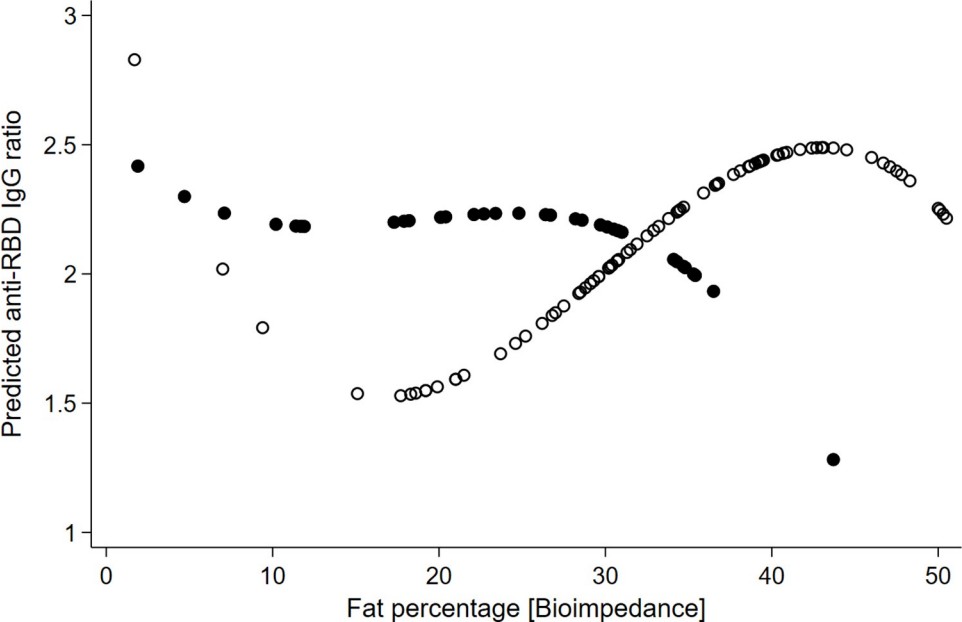

**Fig 3. Polynomial regression analysis of anti-RBD IgG ratios predicted by body fat percentage.** This figure shows the relationship between anti-RBD IgG ratios and body fat percentage for university students, modeled with a polynomial regression to capture the nuances of this relationship. A 3rd-degree polynomial regression is applied for females and a 2nd-degree polynomial for males, reflecting the differential complexity of their responses. Females are depicted with white circles, and males with black circles.

Their findings revealed higher levels of anti-S1/2 antibodies in individuals with advanced age, type 2 diabetes, hypertension, and a positive correlation with BMI [36]. Furthermore, women exhibited higher levels of anti-RBD IgG antibodies compared to men. The authors emphasized the vulnerability of individuals with underlying health conditions, including obesity, to SARS-CoV-2 infection, a concern that is particularly pronounced in older populations. While our study focuses on a younger cohort, providing valuable insights into this demographic group.

Other populations have reported similar findings. Yamamoto et al. [37], reported sex–associated differences in the relationship between body mass index and SARS-CoV-2 antibody titers following the BNT162b2 vaccine in a study of 2,435 healthcare workers in Japan. A meta-analysis examining antibody responses to COVID-19 vaccinations also indicated a significant association between obesity and reduced antibody response [38]. Nevertheless, the considerable heterogeneity (88%) observed across studies suggests that biological factors, including sex, age, and body fat, play a pivotal role in these outcomes.

Tailoring vaccination plans based on an individual's characteristics may enhance vaccine effectiveness. Addressing sex-specific and body fat-related factors in public health interventions can reduce infection rates. Considering the social determinants in the U.S.-Mexico border region, the information provided in this study can help in programs to educate individuals about their susceptibility to infections.

One strength of the study was that the interaction terms captured the combined effect of how body fat percentage and sex jointly influence the antibody response to a vaccine. Another was evaluating multiple models, including first—to third-degree polynomial regressions (nonlinear models), which allowed for a comprehensive data analysis. This approach helped determine the best model that fits the data, ensuring robust and reliable results.

While the present study yields valuable insights, it is important to acknowledge several potential limitations. Variations in immune responses across different age groups, the influence of genetic factors, and the impact of social determinants can introduce complexities that our study may not fully capture. Moreover, it's essential to recognize that the study's cross-sectional design allows for identifying associations but does not establish causality. These considerations emphasize the need for caution in generalizing the findings and highlight avenues for further research.

## Conclusion

This study provides novel insights into the response of anti-RBD IgG antibodies to vaccination in a young cohort. The findings reveal a complex relationship between sex and body fat percentage, depicted by a third-degree polynomial curve (Fig 3). This emphasizes the intricate interplay between body fat and the immune response to vaccines and accentuates the importance of considering sex-specific factors, especially among younger individuals. The complexity of these interactions supports the need for further studies that explicitly include the analysis of sex and fat percentage. Comprehensive knowledge of distinct characteristics and immunological responses helps to understand social and biological dynamics for tailoring vaccination strategies, optimizing public health interventions, and reducing health disparities.

## Supporting information

**S1 Checklist. Inclusivity in global research.**
(DOCX)

**S1 Data. Dataset.**
(XLSX)

## Acknowledgments

Preliminary results of this study were presented and recognized as the best oral clinical presentation at the UTRGV School of Medicine's 5 th Annual Research Symposium, 2021, Mission, TX. The study was supported by a grant to ALPT from the Convocatoria 2021–01: Impulso a la Investigación Científica y de Tecnología Aplicada, COTACyT grant number: COTACYT-2021-01-23. The authors acknowledge the generous support the Universidad Mexico Americana del Norte, the Universidad Autónoma de Tamaulipas for supporting ESFUERSO sharing spaces, personnel, laboratory facilities, and the enthusiastic participation of alumni, staff, and faculty. The funders had no role in study design, data collection and analysis, decision to publish or preparation of the manuscript.

## Author Contributions

**Conceptualization:** Lucia M. Perez-Navarro, Juan C. Lopez-Alvarenga.

**Formal analysis:** Lucia M. Perez-Navarro, Yoscelina Estrella Martinez-Lopez, Juan C. Lopez-Alvarenga.

**Funding acquisition:** Adriana L. Perales-Torres, Esperanza M. Garcia-Oropesa, Octelina Castillo, Juan C. Lopez-Alvarenga.

**Investigation:** Adriana L. Perales-Torres, Esperanza M. Garcia-Oropesa, Marisol Rosas, Octelina Castillo, Laura Ramirez-Quintanilla, Jacquelynne Cervantes, Edda Sciutto.

**Methodology:** Jacquelynne Cervantes, Edda Sciutto, Juan C. Lopez-Alvarenga.

**Project administration:** Adriana L. Perales-Torres, Esperanza M. Garcia-Oropesa, Marisol Rosas, Octelina Castillo, Laura Ramirez-Quintanilla, Claudia X. Munguia Cisneros, Carlos Ramirez-Pfeiffer.

**Resources:** Adriana L. Perales-Torres, Esperanza M. Garcia-Oropesa, Marisol Rosas, Octelina Castillo, Laura Ramirez-Quintanilla, Jacquelynne Cervantes, Edda Sciutto, Claudia X. Munguia Cisneros, Carlos Ramirez-Pfeiffer.

**Supervision:** Adriana L. Perales-Torres, Claudia X. Munguia Cisneros, Carlos Ramirez-Pfeiffer, Juan C. Lopez-Alvarenga.

**Validation:** Adriana L. Perales-Torres, Esperanza M. Garcia-Oropesa, Marisol Rosas, Octelina Castillo, Laura Ramirez-Quintanilla.

**Writing – original draft:** Juan C. Lopez-Alvarenga.

**Writing – review & editing:** Adriana L. Perales-Torres, Lucia M. Perez-Navarro, Alvaro Diaz-Badillo, Yoscelina Estrella Martinez-Lopez, Octelina Castillo, Jacquelynne Cervantes, Edda Sciutto, Leonel Vela, Beatriz Tapia, Juan C. Lopez-Alvarenga.

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
