## [Decision Letter · Decision Letter 0]

30 Jan 2024

PGPH-D-23-02208

Adiposity and Sex Influence on SARS-CoV-2 Antibody Response in University Students. An ESFUERSO cross-sectional study.

Dear Dr. Lopez-Alvarenga,

Thank you for submitting your manuscript to PLOS Global Public Health. After careful consideration, we feel that it has merit but does not fully meet PLOS Global Public Health’s publication criteria as it currently stands. Therefore, we invite you to submit a revised version of the manuscript that addresses the points raised during the review process.

We look forward to receiving your revised manuscript.

Kind regards,

Abram L. Wagner, PhD, MPH

Academic Editor

Journal Requirements:

Additional Editor Comments (if provided):

Reviewers' comments:

Reviewer's Responses to Questions

**Comments to the Author**

1. Does this manuscript meet PLOS Global Public Health’s publication criteria? Is the manuscript technically sound, and do the data support the conclusions? The manuscript must describe methodologically and ethically rigorous research with conclusions that are appropriately drawn based on the data presented.

Reviewer #1: Yes

Reviewer #2: Yes

Reviewer #3: Partly

Reviewer #4: Yes

Reviewer #5: Partly

2. Has the statistical analysis been performed appropriately and rigorously?

Reviewer #1: Yes

Reviewer #2: Yes

Reviewer #3: I don't know

Reviewer #4: Yes

Reviewer #5: No

3. Have the authors made all data underlying the findings in their manuscript fully available (please refer to the Data Availability Statement at the start of the manuscript PDF file)?

Reviewer #1: Yes

Reviewer #2: Yes

Reviewer #3: Yes

Reviewer #4: Yes

Reviewer #5: Yes

4. Is the manuscript presented in an intelligible fashion and written in standard English?

Reviewer #1: Yes

Reviewer #2: Yes

Reviewer #3: Yes

Reviewer #4: Yes

Reviewer #5: Yes

5. Review Comments to the Author

Reviewer #1: The study explored an important area of significant public health interest. It adds important findings to the already existing information and opens a window for more research. The methodology is explicit and supported by rigorous statistical analysis. Ethical issues have been taken in to consideration and the language used is explicit. The main findings are well presented. Limitations of the study have been highlighted for appropriate interpretation of results.

Reviewer #2: Manuscript is good, clear and recommend to publish with minor revision. First of all, there is need to describe more preciously figure 1 and figure 2, or present those figures in tables.

All other section are clear.

Reviewer #3: The major strengths of the paper are the collection of anthropometric and lab data, and the relevance of the topic by both disease burden and underserved population, as well as to current policy debates. The major weaknesses of the paper are mostly related to drafting. The paper requires a more substantive background section to set up the literature that appears in the otherwise well-done discussion section and to motivate the study. The paper also requires a lot more detail to describe the methods so that they can be assessed by reviewers and reproduced by others.

I think the paper is potentially an important contribution to the field but requires additional writing particularly in the background and methods sections, and at least one additional table.

My main comments are below, followed by a few line-by-line substantive comments, and then some minor copyediting comments.

Main comments:

The background section is a little sparse. At the moment, I’m a little unclear as to how this study was motivated. It’s also not clear what your main independent variable was before you ran the analysis. Was it the SISCO measures? Or was this an exploratory study to see what factors account for immune response? If so, it would be helpful to state that.

How and why did you choose the factors to analyze? You’ve provided some justification for assigned sex at birth and adiposity in the background, but not for other covariates or for your main outcome. Though it’s not clear to me that you’re actually building models for sex and adiposity that would adjust for the right confounders. Does the literature point the way towards any covariates that would be important to include in measuring the effects of adiposity and sex on immune response? Is there literature on other vaccines and interactions between adiposity and sex?

Much more information is needed in the methods section on how all of your measures were operationalized. E.g. you note a “metabolic risk group” in 173-174, but didn’t describe in methods how you created this group. How was vaccination assessed? Was it self-report, administrative records, medical records, something else? How were body fat percentage, weight circumference, neck circumference, and blood pressure assessed? It’s important to describe for all those measures, but particularly for body fat percentage as it’s central to your analysis.

Every model that you ran should be described in the methods section, and ideally there should be a table with your results (including the non-significant ones). The way I understand it, you ran a separate model for each of the 6 factors listed at the bottom of Table 1 plus another for food consumption, though that number may be a lot more since it appears as if you tested different models for interactions separately. Or was it just one model with all of the factors loaded in? Describing it in the methods section would greatly clarify how many models for the reader, which is important because depending on how many models you ran, there starts to become an increased risk that that many tests of significance may result in something statistically significant just due to the sheer volume of tests.

It’s still not clear to me which model you ran includes the interaction term for fat percentage and sex.

Table 1: I see food consumption as something that was measured and analyzed (195). How that was measured and operationalized should be described in methods, and should be included in table 1.

The dataset was not available at the provided link (it may have required a UT:RGV login)

There’s not enough information in methods currently to assess whether the results back up the discussion points.

216-248 is well-written and highly relevant. These studies should have been mentioned in the background to establish the literature and motivate the current study.

Line-by-line comments:

113-114: Do you mean the parents of the students? Or the initial cohort?

114: Why were the students contacted due to the pandemic? Why was it this subset in particular (random, administrative reasons, etc.?) and not the entire cohort?

138: might be useful to include the distance and method of transport and/or summarize the sample preservation procedure during transport

161/165: Is it 114 students or 108?

168-169: Unclear what you mean by “significantly” here, was it statistically significant? Or did they meet some pre-defined threshold?

167: How was sex assignment determined? Was it self-report or administrative data?

184: Describe how to interpret this VIF for readers who may be unfamiliar with VIF.

186: Given the limitations of this study, causality can’t be established here, so be cautious of using “explaining” to describe this relationship.

Minor copyediting comments:

58: unsure what “adeptly” is referring to here

100: Needs a citation

151: needs to be a sentence

151-162: Could use some minor copyediting

167: “female students” rather than "females"

Reviewer #4: General comments:

Well articulated molecular level understudy of vaccine, especially COVID-19 vaccines which was globally used in curbing the pandemic.

This study underscores multiple factors that could affect vaccine take, and can be extrapolated to other vaccines especially childhood RI vaccines looking at familial, environmental, dietary and other domains for further development of vaccines.

Studying younger age-groups as in this study helps to target larger percentage of the population pyramid and hence large vulnerable groups that are our future technocrats and leaders, therefore helping humanity in essence.

Suggestions:

Title:

Adiposity and sex influence in vaccinated university students on SARS COV-2 Antibody response: An ESFUERSO cross- sectional study.

Key words:

COVID-19 vaccines; sex; adiposity; immune response; regression analysis; anti-RBD

Abstract: ok

Introduction: ok

Methods:

Line 137-these aliquots were subsequently

Statistical analysis:

Line 151- with percentages for count variables

Results:

Line 169-conditions were identified

Line171- only 3(3%) of the students

Line 189- the interaction remained in spite of the vaccine types.

Discussion: ok

References:

Other COVID-19 vaccine studies in younger age groups globally could be relevant?.....add a few please

Reviewer #5: Title:

Adiposity and Sex Influence on SARS-CoV-2 Antibody Response in University Students. An ESFUERSO cross-sectional study.

Abstract

Line 66: “… this trend was …” => which trend?

Introduction

Line 76: Insert references after the word conducted.

Line 80: Why reference comes after a dot (.)? => .(1) should read (1). Please correct this in the whole document.

Line 87: “BMI > 40 kg/m2” should read “BMI > 40 kg/m2”.

Line 90: If ESFUERSO is an abbreviation, can it be explained in full here?

Line 88: Insert reference after the word responses.

Lines 91-92: “… which recruited first-year college students living in Reynosa prior the

COVID-19 pandemic in 2018”. => this sentence should be part of methods section, not introduction.

Lines 92-99: “This group of students reported … psychological stress”. This paragraph looks like results rather than introduction. Can this be clarified?

Lines 99-100: “Vaccination 99 efforts began in 2021, employing novel mRNA vaccine used in veterinary science for three decades”. => Did authors do vaccination as part of this study (method)? If not, then this sentence needs reference if it comes from the literature.

Lines 102-103: “The aim of this study was …” => start this as a new paragraph.

Lines 105-106: “The study provides insights in the immune response and the potential implications in the context of COVID-19”. => This sentence should come later in the results or discussion.

Comment for the introduction: This section of the manuscript is too short, less than one page is not sufficient. There is no clear rational of the study described here. Authors need to screen further literature to make this section rich and explain why this study is important and/or needed.

Methods

Line 111: Better to avoid confusion: is the ESFUERSO study the one that is being published in this manuscript or different? How sample size was calculated? 500? 116? 108? Better to stick on sample size for your manuscript and explain how you got this figure. Did the authors check the power for this sample size to be representative of the studied population? A subset of 116, was it for what (survey, blood collection or both)?

Line 110: Based on the content of this paragraph, this title should be: Study sample, design, and data collection. Please describe the study design for your research (it seems to be a quantitative clinical study and comparative, I guess). Did the authors respect GCP procedures? Were there SOP and training of study staff done? What was done for the 

---

## [Decision Letter · Decision Letter 1]

10 May 2024

PGPH-D-23-02208R1

Influence of Adiposity and Sex on SARS-CoV-2 Antibody Response in Vaccinated University Students: A Cross-Sectional ESFUERSO study .

Dear Dr. Lopez-Alvarenga,

Thank you for submitting your manuscript to PLOS Global Public Health. After careful consideration, we feel that it has merit but does not fully meet PLOS Global Public Health’s publication criteria as it currently stands. Therefore, we invite you to submit a revised version of the manuscript that addresses the points raised during the review process.

You made good progress in responding to previous reviewer comments. Reviewer 3 still has some outstanding comments. A lot of this is just to make the organization of your paper a bit tighter. If there can be a clearer thread that is balanced on your study aims (last paragraph of intro) - with the intro building up to it, and the methods explaining what you do as a result, I think this would be helpful.

We look forward to receiving your revised manuscript.

Kind regards,

Abram L. Wagner, PhD, MPH

Academic Editor

Journal Requirements:

1. We have noticed that you have uploaded Supporting Information files, but you have not included a list of legends. Please add a full list of legends for your Supporting Information files after the references list.

Additional Editor Comments (if provided):

Reviewers' comments:

Reviewer's Responses to Questions

**Comments to the Author**

1. If the authors have adequately addressed your comments raised in a previous round of review and you feel that this manuscript is now acceptable for publication, you may indicate that here to bypass the “Comments to the Author” section, enter your conflict of interest statement in the “Confidential to Editor” section, and submit your "Accept" recommendation.

Reviewer #2: (No Response)

Reviewer #3: (No Response)

Reviewer #5: (No Response)

2. Does this manuscript meet PLOS Global Public Health’s publication criteria? Is the manuscript technically sound, and do the data support the conclusions? The manuscript must describe methodologically and ethically rigorous research with conclusions that are appropriately drawn based on the data presented.

Reviewer #2: Yes

Reviewer #3: Partly

Reviewer #5: Partly

3. Has the statistical analysis been performed appropriately and rigorously?

Reviewer #2: Yes

Reviewer #3: Yes

Reviewer #5: Yes

4. Have the authors made all data underlying the findings in their manuscript fully available (please refer to the Data Availability Statement at the start of the manuscript PDF file)?

Reviewer #2: Yes

Reviewer #3: Yes

Reviewer #5: Yes

5. Is the manuscript presented in an intelligible fashion and written in standard English?

Reviewer #2: Yes

Reviewer #3: Yes

Reviewer #5: Yes

6. Review Comments to the Author

Reviewer #2: After revision article is good and clearly described.

Reviewer #3: Re-Review for Perales-Torres et al.

Introduction

The introduction is improved. However, the whole paper would benefit from a conceptual diagram (even a simple one), as well as an objective statement, that connects some of the concepts in the introduction (that you added this round) to your study methods. In your response to reviewer 5 you mentioned that you kept the introduction short to keep things concise: I don’t think it’s about literal length per se, so much as you still have not motivated the study yet in this round of revisions. Right now, thinking from the perspective of a naïve reader, it’s not clear why you selected the methods you did because the research question and how you intend to answer that research question are still not clear. E.g. It is not clear to me why you evaluate *these seven models*? Why does a reader want to look at stress as measured by SISCO here? Why should a reader want to know about the effects of your independent variables on RBD? That’s why it’s not clear from the text whether this paper is exploratory analysis—something I mentioned last round—while your response to reviewers makes it clear that you do have specific hypotheses (and which you begin to hint at this round in the introduction). If you have hypotheses, *make them explicit* to the reader in an objectives statement. A conceptual diagram, with the components explained and referenced in the text of the Introduction, will also help you make explicit what exactly you’re operationalizing using your measures and why. Otherwise, there is not enough information for reviewers or readers to assess the *internal validity* of your study. Your response to reviewers (both to me and to reviewer 5) explains this relatively well, that level of explicit explanation should be in the paper itself.

89-90: needs a reference (unless it’s meant to refer to citation (7), in which case the sentence should be edited for clarity).

99: needs something at the beginning of the sentence to make clear that you’re referring to widespread vaccination efforts, rather than an activity that your study carried out. E.g. “Country-wide vaccination efforts…”

104: needs a citation for the protective epitope. And needs more background explanation + references here to explain why it’s important to study RBD and IgG and their ratio.

Methods

Overall, methods and its relationship to the introduction is somewhat improved since last draft, enabling some evaluation of the statistical methods.

The Study Sample section is much improved especially regarding sample size and how that number was arrived at.

There is still no necessary detail on the method by which many key anthropometric data were collected. E.g. was body fat percentage by caliper? (It seems maybe bioimpedance? according to Figures 1 and 2, but should be in the text). The authors need to go through the paper with the perspective of a reader trying to understand the methods used.

There is still not enough information in the statistical analysis section on your regression analyses. You need to describe each of the 7 models (even if just briefly) more than merely mentioning that there are 7 in 175. You also need to state what you consider your threshold for statistical significance.

125-127: The added explanations of SISCO and what you meant by metabolic risk score are very helpful to the reader.

139-141: Is effective neutralization >1 or <1? From context of your use of “positive anti-RBD”, I guess that it’s >1, but you need to make this explicit for the reader.

143-148: In terms of logical flow of section order, the ethics statement may fit better if it came right after the “study sample” section.

154-158: This is more a set of data measures, rather than statistical methods, and should be moved to the measures section.

164-174: it is not necessary to go into this great of detail about VIF in the methods section. The much better explanation in 200-202 is all that is needed for a reader unfamiliar with VIF to be able to interpret VIF.

Results

196: Table 1: Your measures of positive anti-RBD and metabolic risk score should be in Table 1 as well.

196: Table 1: Food factors show up in Table 2 (as it should), and should show up here in Table 1 as well, as stated in my last round of suggestions. Again, this goes back to the importance of motivating the study well in the Introduction and making explicit your hypotheses. Why was food something you considered, even if it ended up not significant?

Table 2: I appreciate the addition of Table 2, in which you appear to report all results, and not just the significant ones. Table 2 makes the paper much more cohesive.

Table 2: 95% CI would be much more interpretable to a reader than standard errors.

200-202: this is a good added explanation of how a reader can interpret your VIF.

Discussion

Overall, the findings in the discussion section seem consistent with the newly reported results in table 2.

238-263: as mentioned in the last round, these sources should be part of your introduction to help motivate the study, since they would presumably inform the hypotheses that you’re testing.

259: meta-analysis should not be capitalized.

265: your study is on sex differences, not gender differences.

Reviewer #5: Title: Influence of Adiposity and Sex on SARS-CoV-2 Antibody Response in Vaccinated

University Students: A Cross-Sectional ESFUERSO study.

Abstract

OK

Introduction

Lines 95-98: This section is okay but should be part of the methodology rather than introduction.

Lines 103-105: This paragraph gives a good aim of the study. Please explain why you insist on the border between US and Mexico. Do you have a specific scientific reason why this region was selected for your study. In addition, please add specific research questions just after the aim.

I would suggest that authors add a single paragraph explaining the rationale for this study (why is it important).

Methods

Line 113: “a subset of 116 students was contacted…” => better to write “a subset of 116 students were contacted…”.

Lines 113-116: A total of 116 students contacted but 108 gave blood samples. What happened to the remaining 8 students? Did they decline to give consent for study? Please clarify.

Is it the same number (108) that attended the survey questionnaire? Or 116? If the study targeted 500 participants and only 108 were included, did the authors check if this figure sufficient to allow them extrapolate their findings?

Part of the explanations of sample size are interesting to be included in the manuscript.

Line 142: It is important that authors add a brief paragraph about study design. It is seen that is clinical and prospective study using both survey and lab tests. It is better that this is written in the manuscript; people should not guess. Add also here information about GCP and safety as explained in the answer to reviewer (it should be in the manuscript).

Can authors add a paragraph showing the univariate and bivariate analysis. The answers to reviewers have good information than the manuscript itself.

Ethics

Line 143-148: Please add more details here on the fact that participants got sufficient explanation about benefits and risks for the study and they were free to withdraw from the study any time.

Add a title of “Data presentation” under which you explain how your data are presented in this manuscript: graphs (bar chart, trend lines, pie chart, …), table, text, etc.

Results

I have difficult to understand the way results are presented. I leave this to another reviewer.

Discussion

Very good discussion. However, we don’t see statistical test support the statements (CI, p value, etc.).

Conclusion

Line 276: Can you write conclusion as a title as you did for Introduction, Methods, Results, and Discussion?

7. PLOS authors have the option to publish the peer review history of their article (what does this mean?). If published, this will include your full peer review and any attached files.

**Do you want your identity to be public for this peer review?** For information about this choice, including consent withdrawal, please see our Privacy Policy.

Reviewer #2: No

Reviewer #3: No

Reviewer #5: No

---

## [Decision Letter · Decision Letter 2]

5 Jul 2024

Influence of Adiposity and Sex on SARS-CoV-2 Antibody Response in Vaccinated University Students: A Cross-Sectional ESFUERSO study .

PGPH-D-23-02208R2

Dear Dr. Lopez-Alvarenga,

We are pleased to inform you that your manuscript 'Influence of Adiposity and Sex on SARS-CoV-2 Antibody Response in Vaccinated University Students: A Cross-Sectional ESFUERSO study .' has been provisionally accepted for publication in PLOS Global Public Health.

Best regards,

Abram L. Wagner, PhD, MPH

Academic Editor

Reviewer Comments (if any, and for reference):

Reviewer's Responses to Questions

**Comments to the Author**

1. If the authors have adequately addressed your comments raised in a previous round of review and you feel that this manuscript is now acceptable for publication, you may indicate that here to bypass the “Comments to the Author” section, enter your conflict of interest statement in the “Confidential to Editor” section, and submit your "Accept" recommendation.

Reviewer #3: (No Response)

Reviewer #5: (No Response)

2. Does this manuscript meet PLOS Global Public Health’s publication criteria? Is the manuscript technically sound, and do the data support the conclusions? The manuscript must describe methodologically and ethically rigorous research with conclusions that are appropriately drawn based on the data presented.

Reviewer #3: Yes

Reviewer #5: Partly

3. Has the statistical analysis been performed appropriately and rigorously?

Reviewer #3: Yes

Reviewer #5: Yes

4. Have the authors made all data underlying the findings in their manuscript fully available (please refer to the Data Availability Statement at the start of the manuscript PDF file)?

Reviewer #3: Yes

Reviewer #5: Yes

5. Is the manuscript presented in an intelligible fashion and written in standard English?

Reviewer #3: Yes

Reviewer #5: Yes

6. Review Comments to the Author

Reviewer #3: This version is greatly improved, the authors are to be commended. All major issues have been addressed. Here are my remaining minor comments:

113-116 does not follow very well from the pervious sentence. The sentence would work better immediately following the sentence in line 99 instead.

218: The sentence is a little awkwardly constructed.

Reviewer #5: Title:

Influence of Adiposity and Sex on SARS-CoV-2 Antibody Response in Vaccinated

University Students: A Cross-Sectional ESFUERSO study.

Abstract

OK

Introduction

OK

Methods

Lines 113-116: A total of 116 students contacted but 108 gave blood samples. What happened to the remaining 8 students? Did they decline to give consent for study? Please clarify.

Is it the same number (108) that attended the survey questionnaire? Or 116? If the study targeted 500 participants and only 108 were included, did the authors check if this figure is sufficient to allow them to extrapolate their findings?

I did not find answers to the above comments. Maybe I am not able to understand it. So, I leave it to other reviewers.

Ethics

Line 143-148: Please add more details here on the fact that participants got sufficient explanation about benefits and risks for the study, and they were free to withdraw from the study any time.

I did not see this in the manuscript. Maybe it is not relevant. So, I leave it to other reviewers.

Results

I have difficult to understand the way results are presented. So, I leave this to another reviewer.

Discussion

OK

Conclusion

Line 276: Can you write conclusion as a title as you did for Introduction, Methods, Results, and Discussion?

This was not addressed. Maybe it is not relevant. So, I leave it other reviewers.

7. PLOS authors have the option to publish the peer review history of their article (what does this mean?). If published, this will include your full peer review and any attached files.

**Do you want your identity to be public for this peer review?** For information about this choice, including consent withdrawal, please see our Privacy Policy.

Reviewer #3: No

Reviewer #5: No
